# Estimating Rainfall from Surveillance Audio Based on Parallel Network with Multi-Scale Fusion and Attention Mechanism

**Mingzheng Chen** [1,2,3], **Xing Wang** [1,2,3,4] , **Meizhen Wang** [1,2,3,*] , **Xuejun Liu** [1,2,3], **Yong Wu** [5] **and Xiaochu Wang** [1,2,3]

1   Key Laboratory of Virtual Geographic Environment (Nanjing Normal University), Ministry of Education, Nanjing 210023, China
2   State Key Laboratory Cultivation Base of Geographical Environment Evolution (Jiangsu Province), Nanjing 210023, China
3   Jiangsu Center for Collaborative Innovation in Geographical Information Resource Development and Application, Nanjing 210023, China
4   Department of Geography and Regional Research, University of Vienna, 1010 Vienna, Austria
5   Institute of Geography, Fujian Normal University, Fuzhou 350000, China
*   Correspondence: wangmeizhen@njnu.edu.cn

**Abstract:** Rainfall data have a profound significance for meteorology, climatology, hydrology, and environmental sciences. However, existing rainfall observation methods (including ground-based rain gauges and radar-/satellite-based remote sensing) are not efficient in terms of spatiotemporal resolution and cannot meet the needs of high-resolution application scenarios (urban waterlogging, emergency rescue, etc.). Widespread surveillance cameras have been regarded as alternative rain gauges in existing studies. Surveillance audio, through exploiting their nonstop use to record rainfall acoustic signals, should be considered a type of data source to obtain high-resolution and all-weather data. In this study, a method named parallel neural network based on attention mechanisms and multi-scale fusion (PNNAMMS) is proposed for automatically classifying rainfall levels by surveillance audio. The proposed model employs a parallel dual-channel network with spatial channel extracting the frequency domain correlation, and temporal channel capturing the time-domain continuity of the rainfall sound. Additionally, attention mechanisms are used on the two channels to obtain significant spatiotemporal elements. A multi-scale fusion method was adopted to fuse different scale features in the spatial channel for more robust performance in complex surveillance scenarios. In experiments showed that our method achieved an estimation accuracy of 84.64% for rainfall levels and outperformed previously proposed methods.

**Keywords:** rainfall estimation; surveillance audio; machine learning; multi-scale fusion

## 1. Introduction

Rainfall is a key factor in the water cycle and the most important input to hydrological modeling studies and hydrological calculations. Persistent or short-term heavy rainfall is the main driving phenomenon of runoff mechanisms, especially in urban areas. The small size of the urban catchments and the high spatiotemporal variability of rainfall necessitate the consideration of rainfall at small scales. Hence, rainfall data with a high spatiotemporal resolution are essential for urban hydrological modeling [1–6] At present, rainfall observation methods are primarily either rain gauge-based or remote sensing inversion-based. However, the former only obtain rainfall data at specific observation stations at regular times, while the latter is restricted by the resolution and update cycle, and underperforms in specific scenarios, such as urban waterlogging and emergency rescue. Studies have reported that hydrological applications for urban catchments of the order of 1000 ha require a temporal resolution of approximately 5 min and a spatial resolution of approximately 3 km [1]. Therefore, it is difficult for the existing rainfall observation

methods to obtain high-resolution precipitation data and satisfy the data needs of urban hydrological studies [4,7–9]. In other words, developing a new platform/sensor to achieve high-resolution and low-cost rainfall monitoring is needed. According to a survey by Comparitech, there are approximately 770 million surveillance cameras worldwide [10]. Widespread surveillance cameras have been regarded as alternative rain gauges in existing studies because they continuously record rainfall events, even though they are mainly used to monitor moving objects or/and detect change. In contrast to surveillance video that only plays an important role during daylight to acquire the visual features of rainfall, surveillance audio can be exploited at all hours to report rainfall acoustic signals and complement the existing rainfall observation network. Thus, surveillance audio should be considered as a type of data source to obtain high-resolution, all-weather rainfall data.

Surveillance audio-based rainfall estimation (SARE) can be regarded as a process of sound classification, which consists of feature extraction and classification [11]. Traditional sound classification uses classifiers (e.g., Gaussian mixture modeling [12], hidden Markov model [13], and support vector machine [14]) to classify audio features (e.g., linear prediction coefficient [15], linear predictive cepstral coefficient [16], and mel-frequency cepstral coefficient [MFCC] [17]), and it is difficult to obtain robust and precise results in a complex urban acoustic environment. Recently, with the rapid development of deep learning (DL), an increasing number of acoustic recognition methods based on DL have been proposed and achieved significant results [18–22]. Because of its powerful generalization ability and better robustness, DL provides a reliable solution for audio research. However, due to the specificity of surveillance audio, the combined effects of a complex urban acoustic environment and variable rainfall sound have brought challenges to current DL methods [23]. For these reasons, methodologies developed for common acoustic recognition cannot be directly extrapolated to SARE. Additionally, most of the existing audio-based rainfall sensing adopts a specific acoustic sensor, which suffers from the limitations of high-cost and difficult installation conditions. However, a surveillance camera with a cost-effective sensor has not been extensively studied [11,24,25]. Wang et al., built an automatic rainfall observation system based on surveillance audio-fusing audio features and adopted a basic convolutional neural network (CNN) as the feature extraction structure to obtain the precipitation levels, and achieved good results. Nevertheless, the spatiotemporal representation of rainfall events and the variability of different scales should be considered. Given the particularity of the problem, we present our technical framework that focuses on the change rule of rainfall events in surveillance scenarios to classify rainfall levels.

To realize effective rainfall estimation and fully excavate monitoring resources, we propose a DL algorithm named the parallel network based on attention mechanisms and multi-scale fusion PNNAMMS, which can learn powerful spatiotemporal features from surveillance audio for rainfall level classification. PNNAMMS has a dual-channel architecture, composed of a CNN and a long short-term memory (LSTM) network, to extract the spatial and temporal features of surveillance audio. Second, in light of the different effects on the final results of different features, we add an attention mechanism to give prominence to the key messages. Moreover, to enhance the discriminative power of the model in different scenarios, we propose a multi-scale fusion method to fuse features from different scales in the spatial channel (SC). Third, the features of the dual channels are fused, and the results are predicted. In terms of training and evaluation, the PNNAMMS is trained and assessed on the Rainfall Audio_XZ (RA_XZ) dataset [11]. The experiments results indicate that PNNAMMS achieved considerable accuracy on the RA_XZ dataset, which exceeds those of the existing mainstream DL methods.

This paper's makes the following contributions:

1.  A parallel dual-channel network model called PNNAMMS is proposed for extracting different features of surveillance audio.
2.  A multi-scale fusion block and attention mechanisms are used in the model to better select features. Further, the impact of different multi-scale fusion methods and attention mechanisms on the performance of the model is explored.

3. The results obtained using PNNAMMS to estimate rainfall levels are presented and analyzed.

The rest of this paper is organized as follows. Section 2 presents Methodology. Section 3 describes the experimental setup and evaluation criteria. Section 4 implements and discusses the proposed methodology. Section 5 discusses the different components of the model. Finally, Section 6 presents concluding remarks.

## 2. Methodology

### 2.1. Main Workflow

Raindrops are the basic unit of rainfall. Notably, the surveillance audio of rainfall is regarded as a continuous superposition and combination of many raindrop sounds. Sometimes, other sounds, such as wind and the noise of cars, can be heard within a certain range of perception. For example, Figure 1 shows the soundscape of a rainfall event. On the whole, the waveform and mel spectrogram show that this rainfall event was successive in the time domain and that different rainfall intensities followed different patterns. However, this continuity can be destroyed by the combined effects of noise. Moreover, owing to the complexity of surveillance scenes, noises occur randomly, resulting in feature changes in the audio's time-frequency domain, creating significant challenges to the modeling of rainfall events [24,25]. In summary, surveillance rainfall audio has continuity, but it is vulnerable to noise, which leads to its difficulty in representing rainfall events with a single domain. Accordingly, we attempt in this study to calculate rainfall in terms of spatial and temporal audio representations.

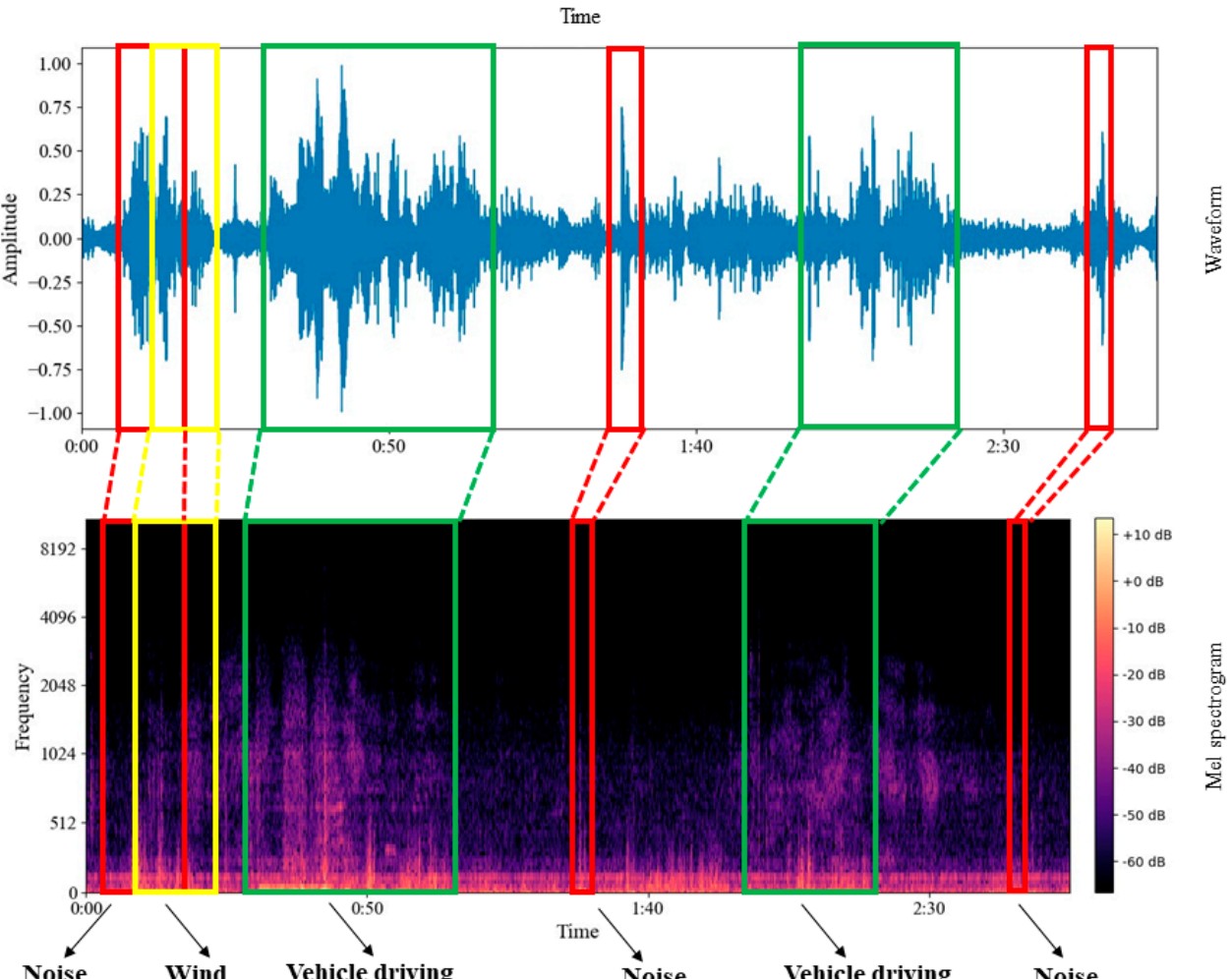

**Figure 1.** Audio schematic of rainfall scenario.

We first designed a general SARE architecture system. As shown in Figure 2, the SARE system is divided into four sections: data processing, feature extraction, model training, and rainfall-level estimation. The data processing refers to splitting the surveillance audio into audio segments according to the specific parameters (audio length and overlap). The feature extraction section extracts the initial acoustic signal of the audio segments. In this study, we focus on model training and rainfall level estimation. Based on previous acoustic analyses, we developed the PNNAMMS data-driven parallel network to extract the spatiotemporal features of rainfall audio and estimate rainfall level (detailed further in Section 2.2).

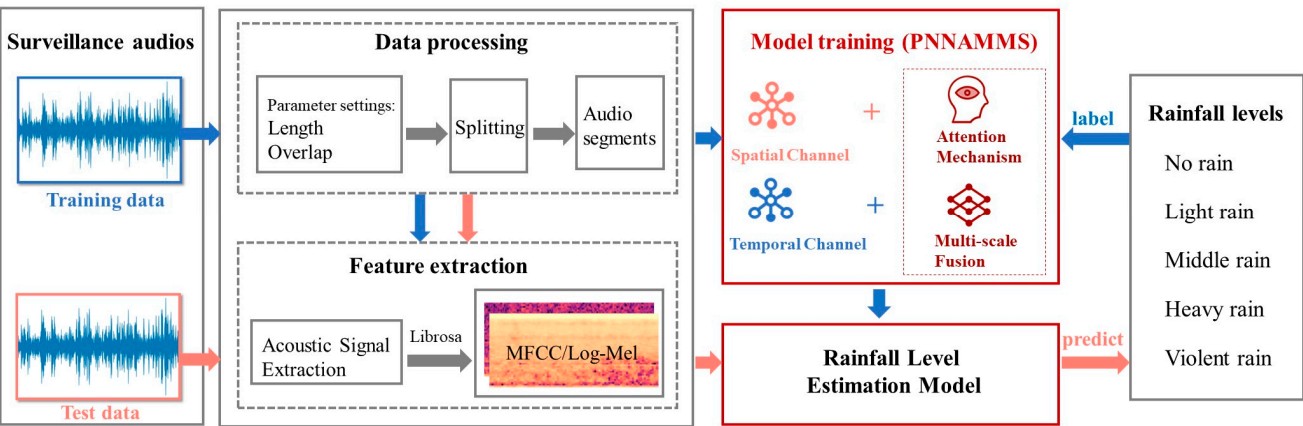

**Figure 2.** Surveillance audio-based rainfall estimation (SARE) architecture system based on the proposed parallel network based on attention mechanisms and multi-scale fusion (PNNAMMS).

### 2.2. PNNAMMS

Figure 3 shows the architecture of the PNNAMMS. The network comprises spatial channel (SC) and temporal channel (TC) sub-networks, a multi-scale fusion block, an attention mechanism, and classification blocks. Specifically, the SC and TC sub-networks constitute a parallel network that captures spatial and temporal representations from surveillance audio signals, respectively. Here, MFCC and Log-mel features were extracted by Librosa [26] and selected as the input to the dual channels. MFCC is the feature parameter with the best representation of the original sound and is used in most application scenarios. Log-mel features combine long- and short-term information representing the temporal distribution of audio events. For parameter settings, we set the MFCCs to 40, and the filter for the Log-mel spectrum to 128. Finally, the generated MFCC feature dimensionality was $40 \times 173$, and the Log-mel feature dimensionality was $128 \times 173$.

To extract more representative features, the following strategies were added to PNNAMMS. First, considering the different effect weights of the final recognition result of different parts of the model, an attention mechanism was added to the parallel network. Moreover, we designed various attention mechanisms for the spatial and temporal channels so that they may adapt to different architectures. Second, high-dimensional features pay more attention to worldwide information and tend to ignore shallow features. However, it is difficult to produce discriminative features from the audio in the light-rain scenario, and, thus, this scenario is easily ignored in the deep network. Therefore, we proposed a multi-scale fusion block to fuse features from different scales to improve the model performance in different scenarios Finally, the spatiotemporal vector is sent to the fully connected layers for classification.

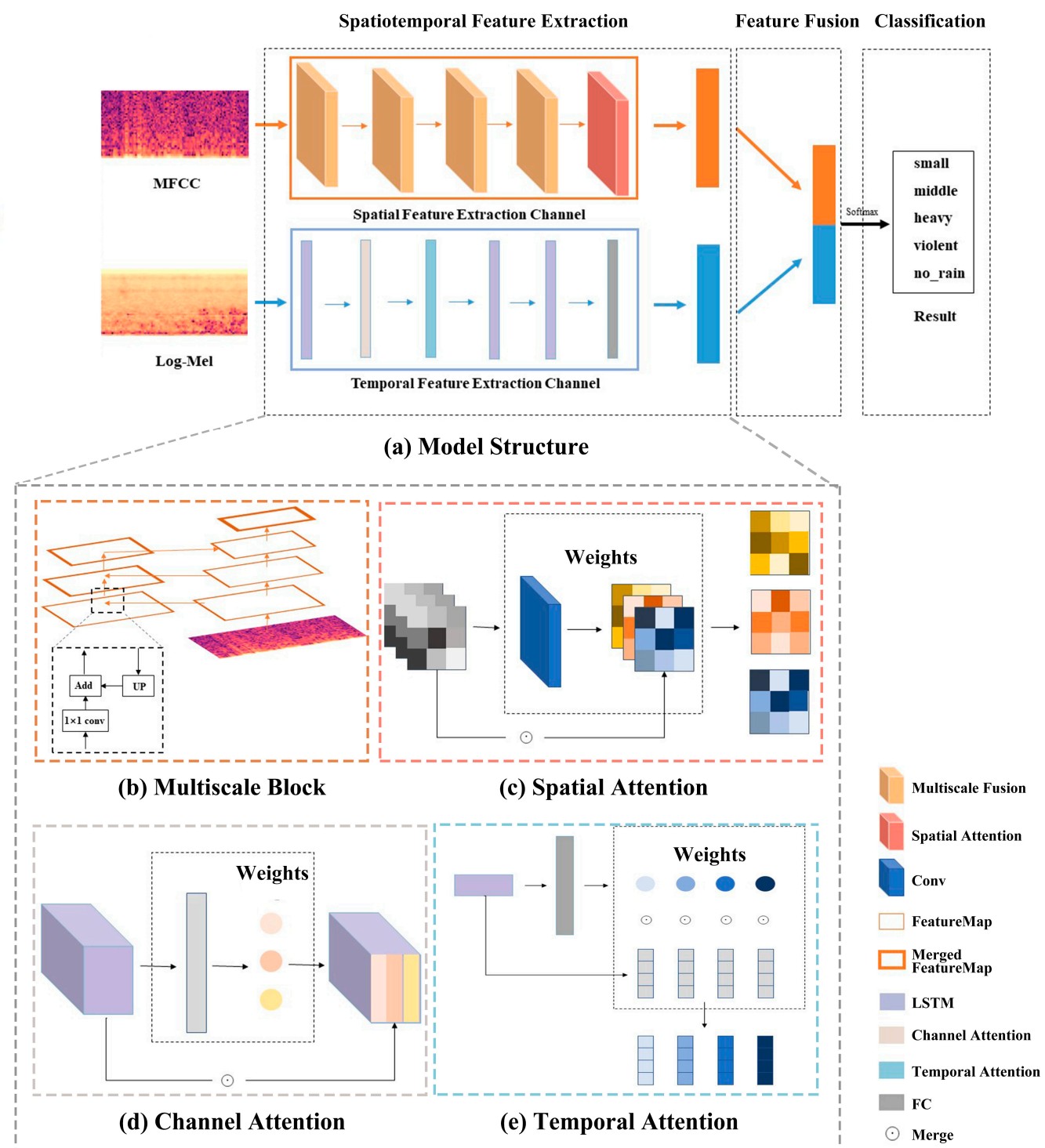

**Figure 3.** PNNAMMS diagram.

### 2.2.1. Parallel Spatiotemporal Network

The MFCC audio features are fed into the SC to learn the spatial correlation. The SC consists of a residual structure, a multi-scale fusion block, and an attention layer (the detailed architecture is shown in Table 1). The residual block makes the training process of the model more convenient and stable through its skip connection, which alleviates the declining accuracy phenomenon when the network goes deep [27]. Therefore, we designed an efficient architecture based on the residual structure to perform MFCC feature extraction. We first pass the features through a residual network to extract audio



features (Figure 4). Then, the weighted multi-scale features are extracted by the multi-scale fusion method described in Section 2.2.2 and the spatial attention mechanism described in Section 2.2.3. The TC takes the Log-mel features as input to capture the temporal continuity of surveillance audio. We adopted the LSTM as the main feature extraction structure and design. Simultaneously, the attention mechanism was added to make the sub-network focus on useful information. Finally, the features of each channel were merged for classification.

**Table 1.** Detailed architecture of PNNAMMS.

| SC Network | | | TC Network | | |
|---|---|---|---|---|---|
| Layer | Output Shape | Setting | Layer | Output Shape | Setting |
| Conv | $(38 \times 171 \times 64)$ | $3 \times 3$, 64 | LSTM | $(128 \times 64)$ | 64, return sequences = True |
| Max pooling | $(19 \times 86 \times 64)$ | $3 \times 3$, stride 2 | | | |
| Residual Block | $(10 \times 43 \times 64)$ | $\begin{pmatrix} 3 \times 3, \ 64 \\ 3 \times 3, \ 64 \end{pmatrix} \times 2$ | Channel attention | $(128 \times 64)$ | |
| Multiscale Block | $(19 \times 86 \times 64)$ | | Temporal attention | $(128 \times 64)$ | |
| Residual Block | $(10 \times 43 \times 128)$ | $\begin{pmatrix} 3 \times 3, \ 128 \\ 3 \times 3, \ 128 \end{pmatrix} \times 2$ | LSTM | $(128 \times 128)$ | 128, return sequences = True |
| Multiscale Block | $(10 \times 43 \times 128)$ | | LSTM | $(64)$ | 64, return sequences = False |
| Residual Block | $(10 \times 43 \times 128)$ | $\begin{pmatrix} 3 \times 3, \ 64 \\ 3 \times 3, \ 64 \end{pmatrix} \times 2$ | FC | $(64)$ | 64 |
| Spatial attention | $(5 \times 22 \times 64)$ | | FC | $(128)$ | 128 |
| Global Average pooling | $(64)$ | | FC | $(64)$ | 64 |
| Concatenate & Classify | | | Result | | |
| Total params | 1,151,855 | | | | |
| Trainable params | 1,147,631 | | | | |

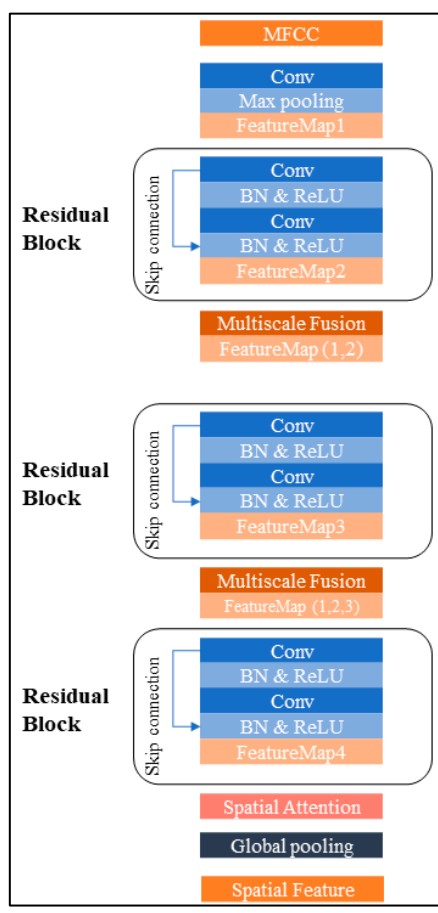

**Figure 4.** Network design of the proposed residual block.

Additionally, to mitigate and suppress overfitting, the following techniques were employed in the PNNAMMS:

(a)  L2 regularization: A regularization penalty term is added to the cost function, reducing the weights by an order of magnitude and mitigating overfitting.
(b)  Batch normalization: A normalization algorithm proposed by Ioffe and Szegedy [28] is used to speed up the convergence and stability of neural networks by normalizing every dimension of each batch of data.
(c)  Spatial dropout: A dropout method proposed by Tompson et al. [29] in the imaging field, which randomly sets some regions to zero, which is effective in image recognition.

### 2.2.2. Multi-Scale Fusion Block

In this study, information on different scales is fused in the SC Network, as shown in Figures 3b and 4. Given an MFCC feature, $X$, of surveillance audio as input, $[X_1,\ X_2,\ X_3,$ and $X_4]$ represents the feature maps of different scales. The deeper feature maps (i.e., $X_3$ and $X_4$) capture more semantic information but possess lower resolution and fewer details. To provide useful supplementary information for classification, the multi-scale block fuses features from different scales through bottom-up, top-down, and lateral connections. The bottom-up method reflects the ordinary forward propagation process. The top-down method up-samples the higher-level feature map, and the lateral connection fuses the feature maps of the two lines. Moreover, the number of feature-map channels is changed using a $1 \times 1$ convolution kernel so that they may be connected with the same feature size. For example, $X_2$ is convolved by a $1 \times 1$ filter to match the number of $X_1$ channels; it is then up-sampled to connect $X_1$ and $X_2$. Afterwards, the fused feature map $[X_1,\ X_2]$ has the information of $X_1$ and $X_2$. Hence, the process of feature fusion can be represented as:

$$X_{fused} = [f(X_3), f(X_2), f(X_1)], \tag{1}$$

where $X_{fused}$ represents the result of feature fusion on the SC, $[\cdot]$ refers to the concatenation of outputs from different layers, $f(\cdot)$ denotes the function used to change the feature channels, and $X_i,\ i = 1, 2, 3$ refers to the $i$th feature map of the channel.

### 2.2.3. Attention Mechanism in PNNAMMS

Attention mechanisms are inspired by the visual system of humans, which tends to concentrate on interesting areas in view while ignoring other less-useful information. This method provides an effective means to improve the efficiency and accuracy of information processing. In this work, we added a spatial attention block (Figure 3c) to the SC network, which is a combination of channel (Figure 3d) and temporal attentions (Figure 3e) in the TC network.

Within the spatial attention block, we followed the design of [30]. First, we concatenated average- and max-pooling to obtain an efficient feature descriptor. Then, a convolution layer was used for weight learning and provided an attention mask. Given a feature map, $F_M(batch\_size,\ height, width)$, the output of the spatial attention block, $F_s$, is formulated as follows:

$$F_s = \sigma(f([AvgPool(F_M); MaxPool(F_M)])) \otimes F_M, \tag{2}$$

where $\otimes$ represents the multiplication of the attention mask and feature map, $\sigma$ refers to the sigmoid function, $AvgPool$ refers to the average-pooling operation, $MaxPool$ refers to the max-pooling operation, and $f$ represents the convolution operation.

The channel attention block focuses on the key information in the channel domain. Considering that the signals from different channels play different roles in the expression of temporal units, the channel attention can focus the TC network on the significant channel. Simultaneously, average- and max-pooling are used to obtain temporal context descriptors. Differing from spatial attention, these are merged and provided with an attention mask through

a sigmoid function. Given a temporal feature, $F_L(batch\_size, time\_steps, lstm\_units)$, the output of the channel attention block, $F_c$, is formulated as follows:

$$F_c = \sigma(f([Add(AvgPool(F_L), MaxPool(F_L))])) \otimes F_L, \tag{3}$$

where $\otimes$ represents the multiplication of the attention mask and feature map, $\sigma$ refers to the sigmoid function, $AvgPool$ refers to the average-pooling operation, $MaxPool$ refers to the max-pooling operation, $Add$ refers to the vector summation operation and $f$ represents the convolution operation.

The temporal attention block focuses on the selection of important temporal contextual information in the time domain. We used a fully-connected layer to generate different weighting factors and applied weight coefficients to each time step. Given a temporal feature, $F_L(batch\_size, time\_steps, lstm\_units)$, the output of the temporal attention block, $F_t$, is formulated as follows:

$$F_t = Softmax((F_L)) \otimes F_L, \tag{4}$$

where $\otimes$ represents the multiplication of the attention mask and feature map.

## 3. Experiments

### 3.1. Dataset

The Rainfall Audio_XZ (RA_XZ) is a DL dataset proposed by Wang et al. for SARE. This dataset comes from Nanjing, China, where cameras were set up at Nanjing Normal University (Figure 5). The RA_XZ dataset takes the structure of the UrbanSound8K [31] dataset as a reference to organize dataset formats, which provide the required information about each audio file (start time, end time, class name, etc.). The labeled surveillance audio comes from a high-precision siphon rainfall meter (ZJC-V, Zhejiang-Hengda Instrument & Meter Co. Ltd., Hangzhou, China), sampled at 22,050 Hz with 64-bit resolution and classified recorded audio to five rainfall intensities [24] (Table 2). In addition, the RA_XZ contains over 10 types of noise data, such as bird calls and car horns, which not only increased data richness while ensuring data accuracy but kept away from the underlying surface that was easy to produce special sounds, such as rain shelter, so that the rainfall audio data could be representative.

**Table 2.** Definitions of rainfall levels.

| Rainfall Level | Rainfall Intensity (r) |
|---|---|
| No rain | r = 0 mm/h |
| Light rain | r ≤ 2.5 mm/h |
| Middle rain | 2.5 mm/h ≤ r ≤10 mm/h |
| Heavy rain | 10 mm/h ≤ r ≤ 25 mm/h |
| Violent rain | r > 2.5 mm/h |

### 3.1.1. Model Training

In the training process of the model, cross-entropy was selected as the loss function. The Adam optimizer ($\beta\_1 = 0.99$, $\beta\_2 = 0.999$) was used [32]. In addition, the initial learning rate was set to 0.001, the batch size to 128, and the epochs to 200, where the learning rate decayed by 10% every 50 steps.

The experiments were conducted on a PC with the Windows 10 operating system and:

(1)  Intel(R) Xeon(R) Bronze 3104 CPU @ 1.70 GHz,
(2)  NVIDIA GeForce GTX 1080 Ti graphics cards,
(3)  32 GB RAM,
(4)  Python 3.6.8, and
(5)  TensorFlow 1.8.0, Keras 2.1.6, and Librosa 0.8.0 libraries [26].

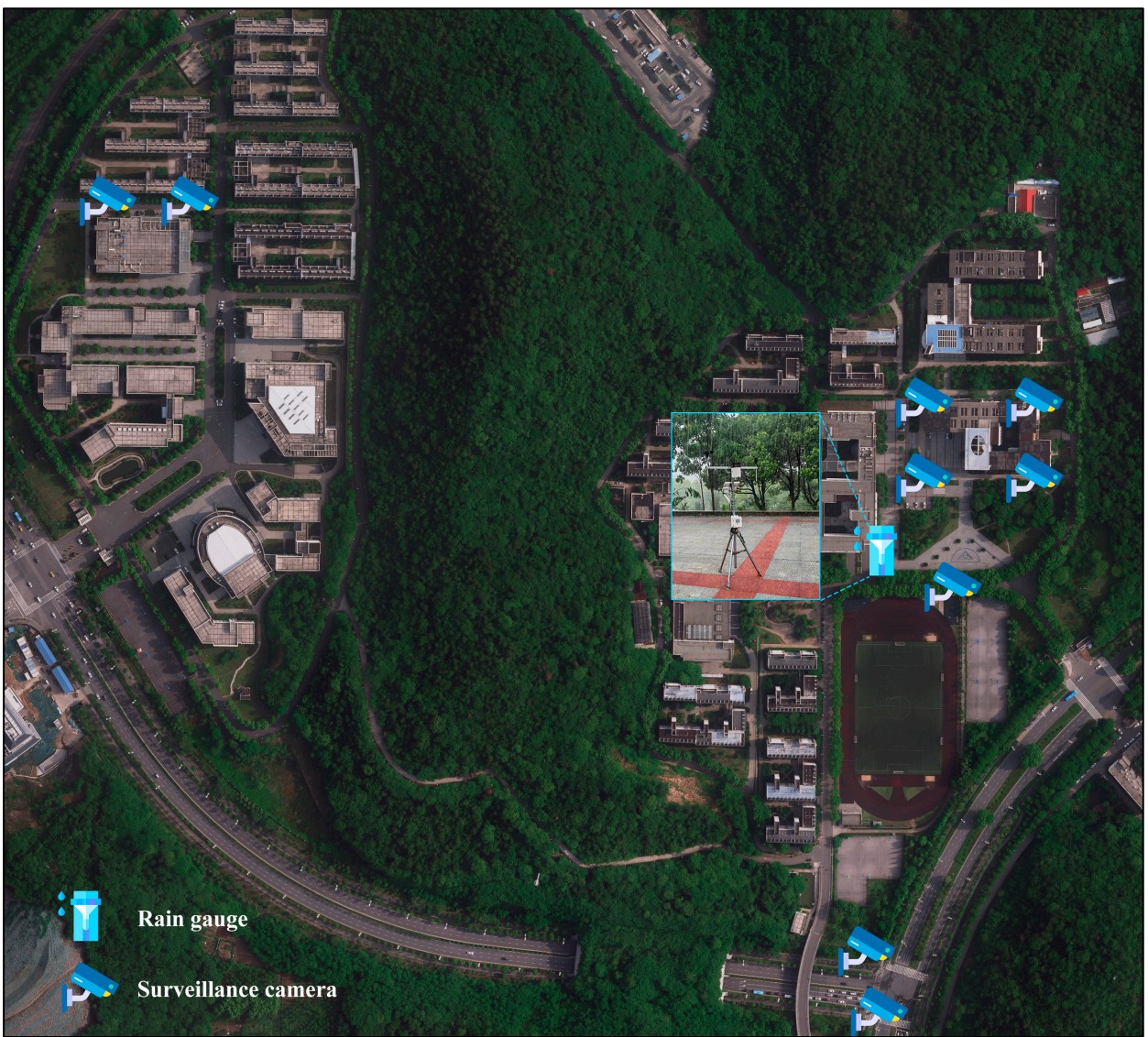

**Figure 5.** Locations of some of the surveillance cameras at Nanjing Normal University.

### 3.1.2. Evaluation Metrics

We utilized four evaluation metrics, *Accuracy*, *F1*, *Precision*, and *Recall*:

$$Accuracy = \frac{C_{corr}}{C_{all}} \tag{5}$$

$$Precision = \frac{TP + TN}{TP + FP + TN + FN} \tag{6}$$

$$Recall = \frac{TP}{TP + FN} \tag{7}$$

$$F1 = \frac{2 \cdot Precision \cdot Recall}{Precision + Recall} \tag{8}$$

*Accuracy* comprised both overall classification accuracy and classification accuracy for each category. $C_{corr}$ represents the number of samples with all correct predictions, and $C_{all}$ represents the total number of all samples. $TP$ represents the number of samples for which the target event was available and determined to be correct, $FP$ represents the number of samples without a target event but determined to have a target event, $TN$ represents the

number of samples with no target events and correct decisions, *FN* represents the number of samples that had a target event but were incorrectly determined. *Precision* indicates the proportion of the sample of all judgments with a positive outcome in terms of the number of correct decisions. *Recall* denotes the proportion of all positive samples that were evaluated to be correct. Each metric has its own focus, and *F*1, which is obtained by averaging the sums of Precision and Recall, provides a balance between them.

## 4. Results

### 4.1. Classification Performance Comparison

To verify the effectiveness of our proposed method, we selected nine state-of-the-art methods for comparison. All these methods were single-feature or multi-feature input methods. FPNet-2D [33], CNN [34], Mel-CNN [35] and Baseline system [36] were designed for mel features, while DSCNN [37] was designed for MFCC features. The multi-feature input methods took multiple features as input. DS-CNN [38] and MRNet [39] were designed for raw and mel features, RACNN [40] was designed for Log-mel and raw features, five-stack CNN [11] was designed for MFCC-Contrast-Chroma features, and PNNAMMS was designed for MFCC and Log-mel features. Multi-feature input models, such as MRNet [39] and PNNAMMS, adopt the idea of using a dual-channel network to extract features and perform classification based on multi-features. In contrast, the five-stack CNN [11] algorithm adapted the method of using fused features as input. In addition, DS-CNN [38] used decision-level fusion to fuse classification results of two network branches and obtains the final result by Dempster–Shafer theory (DS evidence theory).

Table 3 presents results for comparing the performance of the different methods. As Table 3 shows, PNNAMMS achieved an accuracy of 84.64%. The performances of the other nine methods fluctuated between 55.2% and 80.82%. The results indicate that the proposed method is superior to the others. We conclude that our proposed model achieved the best performance on the RA_XZ dataset.

**Table 3.** Performance (Accuracy, Precision, Recall, and F1) of different methods on the RA_XZ dataset.

| Method | Feature | Accuracy | Precision | Recall | F1 |
|---|---|---|---|---|---|
| FPNet-2D | Mel | 0.6173 | 0.6141 | 0.6155 | 0.6056 |
| CNN | Mel | 0.6537 | 0.6511 | 0.6584 | 0.6538 |
| Mel-CNN | Mel | 06383 | 0.6415 | 0.6450 | 0.6385 |
| Baseline system | Mel | 0.6046 | 0.5979 | 0.6056 | 0.5983 |
| DSCNN | MFCC | 0.5520 | 0.5643 | 0.5585 | 0.5602 |
| DS-CNN | Mel&Raw | 0.6271 | 0.6650 | 0.6346 | 0.6105 |
| MRNet | Mel&Raw | 0.7317 | 0.7365 | 0.7342 | 0.7338 |
| 5-stacks CNN | MFCC-C-CH | 0.8082 | 0.8203 | 0.8165 | 0.8178 |
| RACNN | Log-mel&Raw | 0.7721 | 0.7839 | 0.7794 | 0.7765 |
| PNNAMMS | MFCC&Log-mel | 0.8464 | 0.8543 | 0.8569 | 0.8468 |

Table 3 also shows that the average accuracy of the single-input method was 61.38%, while most of the multi-feature models achieved an accuracy of over 70%; this shows that improving multi-feature models are significant. The main reason was that the single-feature input method adopted the idea of using only a simplex feature to represent rainfall audio, which made it difficult to capture deep-level information.

In addition, although the depth of other networks was similar to that of PNNAMMS, most of them extracted features by a simple CNN, resulting in effective spatiotemporal information existing that could not be integrated. In contrast, PNNAMMS not only developed a parallel network to integrate spatiotemporal representation but applied attention mechanisms and multi-scale fusion to obtain significant multi-scale elements. Hence, the proposed method achieved excellent performance.

### 4.2. Rainfall Inversion Validity

Based on the performance in different rainfall scenarios, the classification results of PNNAMMS have been compared with the five best-performing alternative models using a confusion matrix (Figure 6). As Figure 6 shows, PNNAMMS had a classification accuracy for "small" of 77.27%, "middle" of 94.40%, "heavy" of 86.69%, "violent" of 66.66%, and "no_rain" of 97.19%; thus, it achieved excellent classification performance in the categories of "middle", "heavy", "small," and "no_rain". In addition, note that the classification accuracy for "small" of PNNAMMS was higher than that of the other models. Specifically, the accuracy of PNNAMMS was 21.08% higher than that of CNN [34], 6.09% higher than that of five-stack CNN [11], 41.50% higher than that of DS-CNN [38], 18.69% higher than that of MRNet [39], and 12.93% higher than that of RACNN [40]. With the fusion of different scales, our model was more robust than other networks.

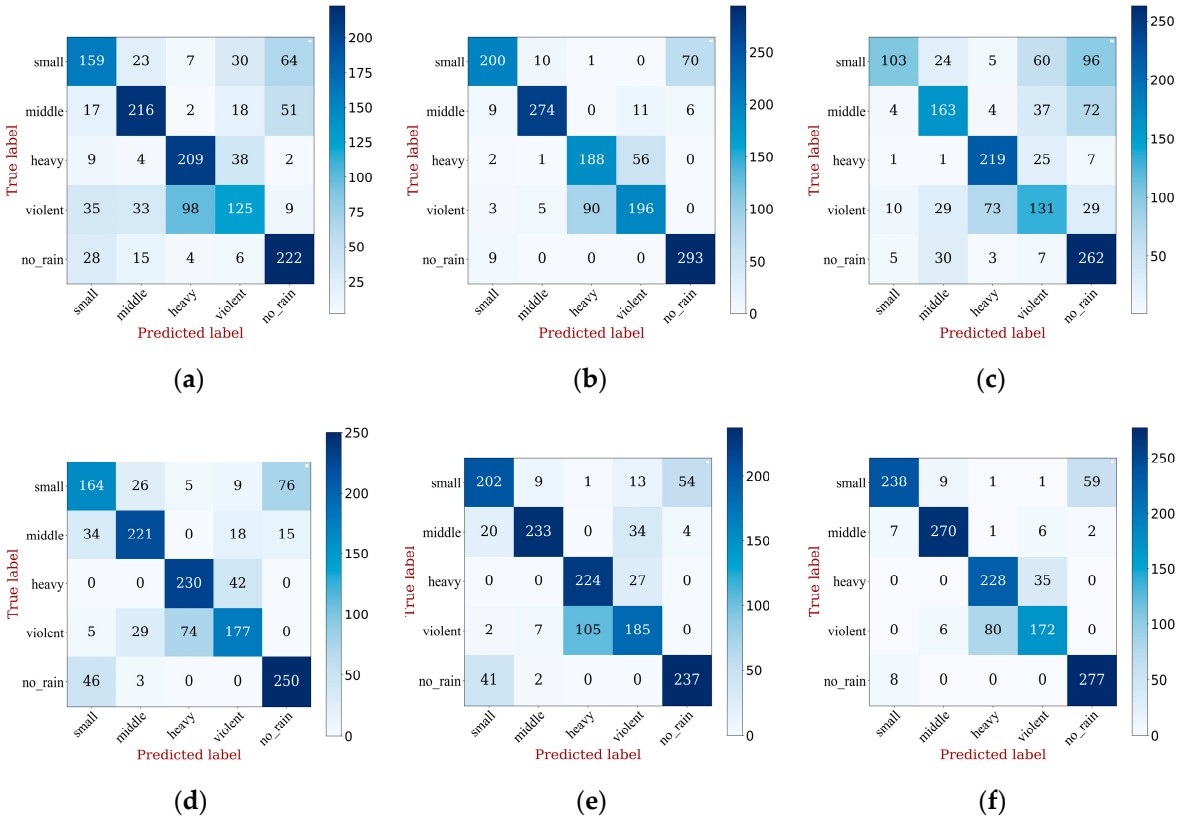

**Figure 6.** Confusion matrixes for different models. (**a**) CNN; (**b**) 5-stack CNN; (**c**) DS-CNN; (**d**) MRNet; (**e**) RACNN; (**f**) PNNAMMS.

However, the "heavy" and "violent" classes were likely to be misjudged. Thus, we randomly selected 100 "heavy" and 100 "violent" audios for Pearson similarity calculation for MFCC and Log-mel features (PPMCC). PPMCC [41] is a measure of the correlation between two sets of vectors; the larger it is, the more similar they are. Table 4 shows the statistical results. Table 4 shows that the features of "heavy" and "violent" were similar, with a mean PPMCC value of 0.9496 for the MFCC features and 0.8938 for the Log-mel features. Moreover, the variance and standard deviation for the MFCC features were 0.0007 and 0.0264, respectively, while for the Log-mel features, it was 0.0016 and 0.0400, respectively. Hence, due to the similarity between "heavy" and "violent," there were certain errors in the discrimination. This was for two main reasons, firstly, the limitations of the hardware of the surveillance camera and the complexity of the surrounding environment, which led to the recording of similar sound features and made it difficult to distinguish

"heavy" from "violent," and secondly, ambient noise, such as wind, which was responsible for classification errors.

**Table 4.** Comparison of PPMCC with different features.

| Index | PPMCC_MFCC | PPMCC_Log-mel |
|---|---|---|
| Average | 0.9496 | 0.8938 |
| Variance | 0.0007 | 0.0016 |
| Standard Deviation | 0.0264 | 0.0400 |

## 5. Discussion

In the experiments, we found that the attention mechanism and multi-scale fusion block had different effects on the accuracy of the model, reflected in the combination of attention mechanisms and hierarchical selection of multi-scale fusion. Therefore, we will discuss the different performances of the model under the aforementioned conditions.

### 5.1. Multi-Scale Feature Fusion Performance Analysis

In the proposed rainfall level classification model, multi-scale fusion played an important role in performance improvement. In this subsection, a subject-dependent experiment was conducted to measure its contribution in different conditions. We used the following method for performance comparisons:

(1) a single baseline network was used for training and evaluating on the RA_XZ dataset;
(2) different combinations of fusion levels were used in the baseline network for training and evaluation (Figure 7).

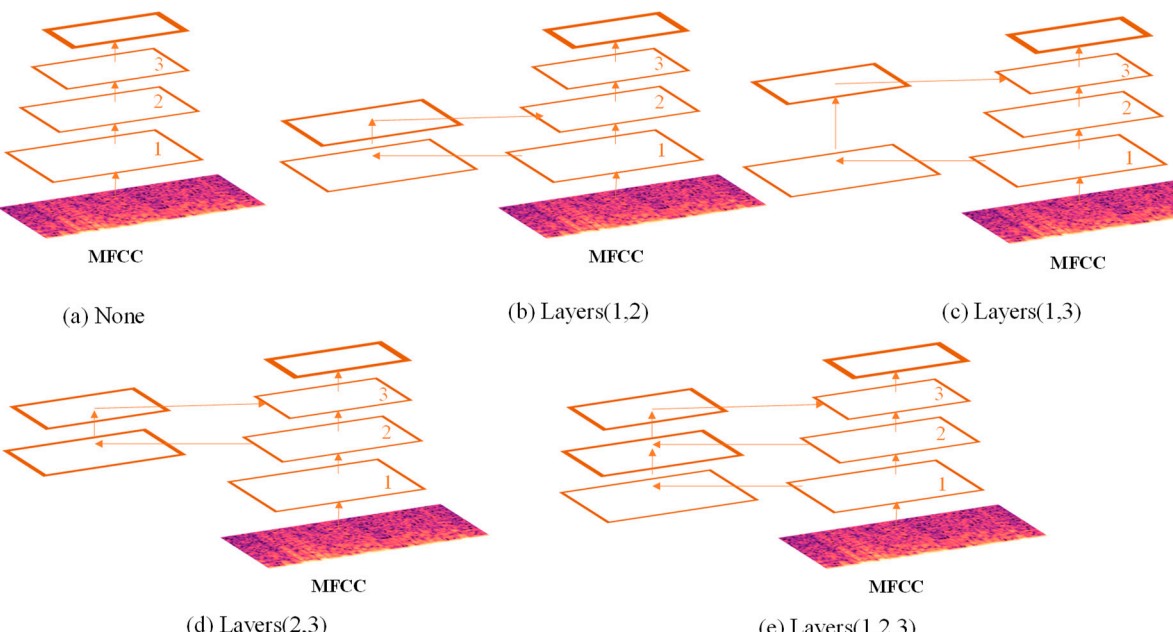

**Figure 7.** Schematic diagrams of different multi-scale structures.

Each baseline network was a parallel neural network based on attention mechanisms (PNNAM).

Table 5 shows the results. We found that although fusion layers might affect the system, the performance of the PNNAM system still surpassed that of most of the other methods; this showed the superiority of PNNAM. In addition, in the case of fusion layers 1, 2, and 3, the model achieved an accuracy of 84.64%, which was superior to those of the other models, due to its powerful ability of feature extraction. However, the fusion model of layers 1 and 2 had the lowest accuracy, which was only 73.28%, due to the confluent feature that

contained too much low-scale information and influenced semantic information. Moreover, the cross-layer fusion models had better performance than others, which demonstrated that the confounding effect between nearby layers had to be reduced by fusing features across layers.

**Table 5.** Comparison of different Multi-level structures in RA_XZ dataset.

| Levels | Accuracy | Precision | Recall | F1 |
|--------|----------|-----------|--------|-----|
| None | 0.7907 | 0.7951 | 0.7945 | 0.7849 |
| Layers(1,2) | 0.7328 | 0.7381 | 0.7335 | 0.73647 |
| Layers(1,3) | 0.8200 | 0.8345 | 0.8279 | 0.8298 |
| Layers(2,3) | 0.7828 | 0.7934 | 0.7856 | 0.7875 |
| Layers(1,2,3) | 0.8464 | 0.8543 | 0.8569 | 0.8468 |

Further, we analyzed the performance of the model in different scenarios (Figure 8). As Figure 8 shows, system performance varied for different fusion layers. For the model-Layers (1,2), the system performance was decreased in most scenarios compared with PNNAM. Specifically, the accuracy values of "small" and "no_rain" were affected. By contrast, some performance criteria were improved with the fusion of Layers (1,3) and Layers (1,2,3). Therefore, adopting the method of multi-scale fusion (cross-layer fusion) was beneficial for the system to be more robust in different scenarios. The main reason was that the feature of light ("small") rain audio was less obvious, and it tended to be ignored after extracting deep-level features of the general neural network. Hence, the fusion of multi-scale features established a more complete feature map and provided better descriptions of the audio.

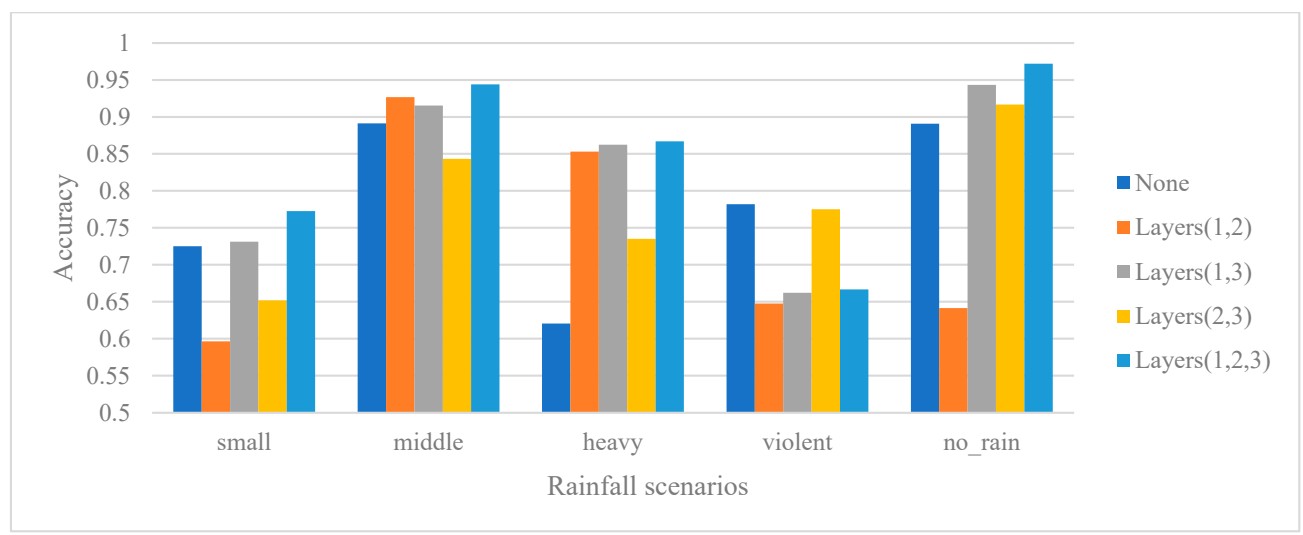

**Figure 8.** Comparison of accuracy for different multi-scale structures.

### 5.2. Performance Analysis of Models with Different Attention Mechanisms

Although we found that the PNNAMMS extracts robust and significant spatiotemporal audio features, the question of what happens to the performance when we use different attention mechanisms remained unanswered. Therefore, we applied the following methods of performance comparison:

(1)  A single baseline network for training and evaluation of the RA_XZ dataset
(2)  Only a single attention mechanism in the baseline network for training and evaluation
(3)  Different combinations of attention mechanisms in the baseline network for training and evaluation

Each baseline network was a parallel neural network based on multi-scale fusion (PNNMS). Table 6 shows that the PNNMS with an attention mechanism performed better than the single one with regard to the accuracy, precision, recall, and F1. The single attention mechanism with a different weighting strategy had a different influence than that of the baseline network. Specifically, the PNNMS with channel attention (C) achieved an accuracy of 81.64%, which is higher than that of the PNNMS alone (none), the PNNMS with spatial attention (S), and the PNNMS with temporal attention (T). Therefore, adopting the method of adding single-channel attention is beneficial for learning essential information.

**Table 6.** Classification performance of different attention mechanisms.

| Attention | Description | Accuracy | Precision | Recall | F1 |
|---|---|---|---|---|---|
| None | None | 0.8114 | 0.8109 | 0.8155 | 0.8148 |
| S | Spatial | 0.8092 | 0.8238 | 0.8125 | 0.8057 |
| T | Temporal | 0.8050 | 0.8146 | 0.8168 | 0.8034 |
| C | Channel | 0.8164 | 0.8345 | 0.8239 | 0.8198 |
| C-S | Channel_Spatial | 0.8071 | 0.8113 | 0.8165 | 0.8143 |
| C-T | Channel_Temporal | 0.8235 | 0.8468 | 0.8265 | 0.8250 |
| S_T | Spatial + Temporal | 0.8242 | 0.9135 | 0.9142 | 0.9130 |
| C-S_T | Channel_Spatial + Temporal | 0.8007 | 0.8188 | 0.8059 | 0.8086 |
| S-C_T | Spatial + Channel_Temporal | 0.8464 | 0.8543 | 0.8569 | 0.8468 |
| C-S_C-T | Channel_Spatial + Channel_Temporal | 0.8200 | 0.8313 | 0.8243 | 0.8240 |

Table 6 shows that not all attention block combinations improve model performance. For the PNNMS with C-S and C-S_T, performance decreased, compared with PNNMS (none). In contrast, other combinations improved model accuracy. Specifically, the PNNMS with S-C_T was the best-performing model with an accuracy of 84.64%. It can be concluded that choosing the optimal attention strategy is important for SARE tasks.

Relative to the experimental results listed in Table 5, Figure 9 shows how different attention mechanisms performed individually in different scenarios. For the single attention block, spatial attention (S) was the best performer for "heavy," but it was the worst for "violent." Temporal attention (T) played an important role in improving "violent," but it had a dampening effect on "heavy." Channel attention (C) had a good impact on most scenarios, indicating that there should be a certain complementarity and contradiction between different attentions.

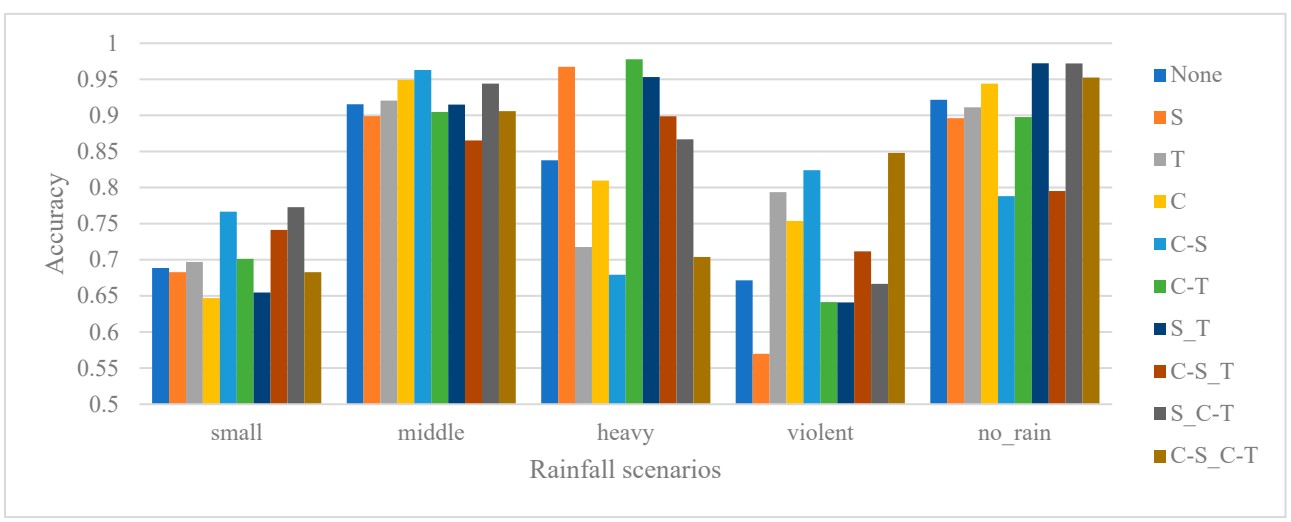

**Figure 9.** Comparison of accuracy for different attention mechanisms (C = Channel Attention, S = Spatial Attention, T = Temporal Attention; "-" denotes the same branch, and "_" denotes different branches).

We also compared the performances of different combinations of attention blocks. As shown in Figure 9, combining attention mechanisms allowed the model to reduce the contradiction between different attentions, which improved performance. For example, although spatial attention (S) was the worst performer for "violent," C-S and C-S_C-T achieved higher accuracy than others, demonstrating that single spatial attention (S) could be improved by the addition of channel (C) and temporal (T) attentions. Additionally, C-T performed well on "small" and "heavy", and S-C_T performed well on "small", "heavy", and "violent"; however, the other scenarios were more inhibited. S-T was somewhat enhanced for each scenario, and S-C_T showed improvements in all scenarios, especially for "small" and "no_rain". C-S_C-T obtained acceptable results, but it was not the best performer and added more parameters to the network. Overall, the best model adopted the S-C_T block, which achieved considerable accuracy in most scenarios. Moreover, the S-C_T block was regarded as a spatial, three-dimensional temporal attention that focused on temporal information in different channels and provided more expressive descriptions of the audio.

## 6. Conclusions

In this study, we proposed a DL framework called PNNAMMS that improves the performance of the SARE system in terms of network architecture and audio features. We showed that when the spatiotemporal network is combined with attention mechanisms and multi-scale fusion blocks, the model, as expected, has higher accuracy and lower errors than when relying on a single CNN only. Having trained and validated the proposed method on surveillance audio data, we then further show that multi-scale and multi-dimensional features can better represent a rainfall event by analyzing the acoustic characteristics of rainfall in urban scenarios. Finally, we demonstrated that surveillance can serve as a trigger for the rapid assessment of rainfall spatiotemporal distribution when an urban environment experiences short spells of heavy rainfall. In this regard, the proposed method gives researchers/decision-makers faster insights into how rainfall effect different areas within an urban environment.

Despite the encouraging results, our study has a number of limitations that should be addressed by future research. First, our study focused exclusively on the effectiveness of surveillance audio to complement the existing rainfall observation network, and cannot answer the question of whether surveillance audio use is better or worse than other sensors. That is, it is currently still a coarse-grained classification of rainfall levels that does not allow for some scenarios that need a quantitative rainfall value. Future research could address this question by constructing a quantitative dataset and exploring a quantitative model. An additional limitation of this study is that it only considers common audio features, such as MFCC, Mel, and Log-Mel. MFCC was selected because of its well-documented excellent performance on a wide range of problems. We did not explore whether there are audio features that are more applicable to rainfall in more scenarios (forests, farmland, sea, etc.). Additional audio features might, however, be explored in future work.

**Author Contributions:** M.C.: Conceptualization, Methodology, Software, Validation, Writing—original draft, Writing—review & editing. X.W. (Xing Wang): Software, Data curation, Writing-review & editing, Supervision. M.W.: Validation, Formal analysis, Supervision, Funding acquisition. X.L.: Conceptualization, Investigation, Project administration. Y.W.: Validation, Visualization. X.W. (Xiaochu Wang): Resources, Data curation. All authors have read and agreed to the published version of the manuscript.

**Funding:** This research was funded by National Key R&D Program of China (2021YFE0112300), National Natural Science Foundation of China (NSFC) (41771420), Special Fund for Public Welfare Scientific Institutions of Fujian Province (2020R11010009-2) and Research program of Jiangsu Hydraulic Research Institute (2020z024).

**Data Availability Statement:** The source codes and data are available for downloading at the link: https://github.com/HouZi64/AMPNN.

**Acknowledgments:** The authors would like to thank the editors and reviewers for their constructive comments, which improved the quality of this paper.

**Conflicts of Interest:** The authors declare no conflict of interest.

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
