# Peer review of "Estimating Rainfall from Surveillance Audio Based on Parallel Network with Multi-Scale Fusion and Attention Mechanism"

_remotesensing, doi:10.3390/rs14225750_

Round 1

Reviewer 1 Report

I’m not an expert in the research field of signal detecting, so I can’t judge how novel this study is in science or methodology, but I personally feel this is a good quality paper in terms of writing. The authors basically proposed a ML-based approach in rainfall classification using data from surveillance audio. I have a few questions when reading this paper:

1.     In Figure 1, how are different noises distinguished? What is Noise in specifical? How do you know when there are overlap between different noises? How do you deal with the signal when the rainfall intensity changes during a period.

2.     Section 3.1, please provide more detailed description about the RA_XZ dataset. It would be straightforward to show the surveillance camera locations in a map.

3.     I would recommend the authors showing more figures that compare your model estimation and label. Maybe some scatter plots would do?

4.     Your approach is developed based on the RA_XZ dataset. How does it perform in other datasets? Is it still good in other datasets?

Author Response

Dear Reviewer,

Thanks for the comments. We gratefully thank you for the time spent making your constructive remarks and useful suggestions, which have significantly raised the quality of the paper and have enabled us to improve the paper. Each suggested revision and comment, brought forward by you, was accurately incorporated and considered. We will be happy to edit the text further, based on your helpful comments.

The text style for revision is as follows:

  • All comments are in black.
  • All responses to the comments are in red.
  • All page and line numbers refer the revised manuscript.
  • All revised contents in the manuscript are highlighted in red.

Response to Reviewer 1 Comments

Point 1: In Figure 1, how are different noises distinguished? What is Noise in specifical? How do you know when there are overlap between different noises? How do you deal with the signal when the rainfall intensity changes during a period.

Response 1: Thank you for your comments, which is highly appreciated. For noise, it is unavoidable in the natural environment. To improve the robustness of model in a noisy scenario, we used deep learning to establish the mapping relationship between surveillance audio and rainfall, as deep learning has a powerful nonlinear feature extraction ability (LeCun Y, Bengio Y, et al., 2015). In this study, we used a parallel architecture to extract rainfall features. Apart from using CNN to extract the rainfall audio features in the frequency domain, we added LSTM to extract the temporal features. Since rainfall audio is continuous and noises are often discrete, we extract the temporal features of rainfall audio, which made it possible to estimate rainfall from contextual information even if the rainfall is covered by noise for a short period.

Point 2: Section 3.1, please provide more detailed description about the RA_XZ dataset. It would be straightforward to show the surveillance camera locations in a map.

Response 2: We apologize for the inadequate description of the RA_XZ dataset and sincerely hope that the information of the RA_XZ dataset is now adequate with this new revision. Firstly, we added the information of data collection location, the definition of rainfall levels and the composition format of the dataset (Please see Line 231-237 and Table 2). Besides, we produced a surveillance camera distribution diagram according to your kind suggestion, which shows the locations of surveillance cameras and rain gauge (Please see Figure 5).

Point 3: I would recommend the authors showing more figures that compare your model estimation and label. Maybe some scatter plots would do?

Response 3: This study focused exclusively on the classification of rainfall levels, the confusion matrix is the most classical way of representing the performance of the model. Besides, the confusion matrix was used in most of the previous studies related to audio classification (Das et al., 2020; Piczak et al., 2015; Karthika et al., 2020; Sharma et al., 2020). Therefore, to facilitate future readers to compare our model with other models, we chose the confusion matrix as the representation of the result. However, scatter plots are very useful for quantitative studies. In the future we will investigate if rainfall can be estimated quantitatively by surveillance audio, then we will use scatter plots. Thanks for your valuable suggestion, it is meaningful for our research.

Point 4: Your approach is developed based on the RA_XZ dataset. How does it perform in other datasets? Is it still good in other datasets?

Response 4: We agree with the reviewer that further experiments on other datasets would be helpful. Unfortunately, RA_XZ is currently the only dataset that can be used to evaluate surveillance audio-based rainfall estimation. There are some environmental sound datasets (e.g. UrbanSound8k, which contains various animal sounds, vehicle sounds, etc.). However, the environment sound datasets did not have rainfall level labels, so they not suitable for the verification of our proposed method.

We sincerely hope that this revised manuscript has addressed all your comments and suggestions. We also asked a professional language editing service to edit the use of English grammar. Once again, thank you very much for your kind comments and suggestions.

References

LeCun Y, Bengio Y, Hinton G, 2015. Deep leaning. Nature, 436-444.

  1. K. Das, A. Ghosh, A. K. Pal, S. Dutta and A. Chakrabarty, 2020. Urban Sound Classification Using Convolutional Neural Network and Long Short Term Memory Based on Multiple Features. 2020 Fourth International Conference On Intelligent Computing in Data Sciences (ICDS). IEEE, 1-9.
  2. J. Piczak, 2015. Environmental sound classification with convolutional neural networks. 2015 IEEE 25th International Workshop on Machine Learning for Signal Processing (MLSP), 1-6.
  3. Karthika and B. Janet, 2020. Deep convolutional network for urbansound classification. Sādhanā 45, 1-8.
  4. Sharma, O. Granmo and M. Goodwin, 2020. Environment Sound Classification Using Multiple Feature Channels and Attention Based Deep Convolutional Neural Network. Interspeech. 1186-1190.

Reviewer 2 Report

Review

The structure of the paper should be improved extensively. The methodology, result and discussion should be separated. In the current version it is all mixed which is hard to follow.

Where is the case study and the coverage? I suppose the performance of the method could alter in another region. Please explain more in detail the case study and the rainfall regime

Introduction:

Generally inetersting topic. But in the first paragaraph the importance of availability to high quality of rainfall data is not clearly shown. The introduction requires better structure and story including the importance of rainfall, the drawbacks of different methods, importance of the current method…

Methodology

2.1 Basic Idea

I do not understand the term Basic idea as a subchapter title. Maybe another wording could be used or generally remove the subchapter title. Here a flowchart could be useful for the reader to understand the general concept of the method, since so many different steps has been implemented.

The chapter Experiments should be included in the Methodology. It could be e.d. 2.2 Datasets, 2.2.1 Data preprocessing

The model training and the Dataprocessing are two different topics.

The result should be shown in the seperate chapter.

Different rainfall type should be defined in the methodology. What does it mean with middle, heavy...

Discussion

line 268: Why again APNN is abbreviated here?

Why Table 4 show the result in discussion but not in the result section? It should be moved.

Conclusion

It is concluded that the rainfall detection has been improved so provide a baseline for the integration of water management. Is it really the fact in comparison to another sensors and methods? Please provide more evidence in the discussion.

Author Response

Dear Reviewer,

Thanks for the comments. We gratefully thank you for the time spent making your constructive remarks and useful suggestions, which have significantly raised the quality of the paper and have enabled us to improve the paper. Each suggested revision and comment, brought forward by you, was accurately incorporated and considered. We will be happy to edit the text further, based on your helpful comments.

The text style for revision is as follows:

  • All comments are in black.
  • All responses to the comments are in red.
  • All page and line numbers refer the revised manuscript.
  • All revised contents in the manuscript are highlighted in red.

Response to Reviewer 2 Comments

 Point 1 : The structure of the paper should be improved extensively. The methodology, result and discussion should be separated. In the current version it is all mixed which is hard to follow.

Response 1: Thanks for your constructive suggestion, we sincerely hope that our logic is now easier to follow with this new revision. We have modified the structure of our paper.

1) Firstly, we revised the first paragraph of the Introduction, describing the importance of high-resolution rainfall data, the role of surveillance audio in rainfall observation networks, and the drawbacks of existing methods.

2) Secondly, we revised the Methodology section, and provided a flowchart to represent the general process of our method.

3) Thirdly, we added more information of dataset, including the definition of rainfall levels, the locations of surveillance cameras, and the label details.

4) Finally, we rewrote the Conclusion section. On the one hand, we described the contribution of our study. On the other hand, we pointed out the limitation of our method and future work.

Point 2 : Where is the case study and the coverage? I suppose the performance of the method could alter in another region. Please explain more in detail the case study and the rainfall regime.

Response 2: The RA_XZ dataset was collected in Nanjing, China. It was recorded in different scenarios in the city (ordinary roads, pavements, residential areas, etc.), therefore we believe that our method/model can be applied in the urban area. To address the reviewer’s concern, we added the information of data collection location, the definition of rainfall levels, and the composition format of the dataset (Please see Line 231-237 and Table 2). Besides, we produced a surveillance camera distribution diagram to show the locations of surveillance cameras and rain gauge (Please see Figure 5).

Point 3 (Introduction): Generally interesting topic. But in the first paragraph the importance of availability to high quality of rainfall data is not clearly shown. The introduction requires better structure and story including the importance of rainfall, the drawbacks of different methods, importance of the current method…

Response 3: Thanks for your constructive suggestions. We have revised the first paragraph of the Introduction according to your suggestion.

1) We pointed out the importance of high spatiotemporal resolution rainfall data. Persistent /short-term heavy rainfall is the main driving phenomenon of runoff mechanisms, especially for the urban area. The small size of the urban catchments and high spatiotemporal variability of rainfall oblige us to consider rainfall at small scales. Hence rainfall data with high spatiotemporal resolution are essential for urban hydrological. Please see Line 56-60.

2) We then indicated the requirements of urban hydrology for rainfall data resolution and that existing rainfall observation methods are difficult to meet this requirement. Furthermore, the necessity of developing new rainfall observation methods is proposed. Please see line 64-68.

3) Finally, we demonstrated the important complementary role of surveillance audio to the rainfall observation network, as it can reflect rainfall information and record all-weather data. Please see line 73-76.

Point4 (2.1 Basic Idea): I do not understand the term Basic idea as a subchapter title. Maybe another wording could be used or generally remove the subchapter title. Here a flowchart could be useful for the reader to understand the general concept of the method, since so many different steps has been implemented.

Response 4: We used “Main workflow” instead of “Basic Idea”. At the same time, we provided a flowchart to help readers to get the general concept of the method according to your kind suggestion (Please see Figure 2). We divided the surveillance audio-based rainfall estimation (SARE) system into Data Processing, Feature Extraction, Model training, and Rainfall level estimation. Then we explained related details and the focus of this study(Please see Line 136-140).  

Point 5 (Datasets): The chapter Experiments should be included in the Methodology. It could be e.d. 2.2 Datasets, 2.2.1 Data preprocessing.

Response 5: We reorganized the structure of this paper. The Data processing section in original version has been removed, and its contents were moved to section 2.1 (Please see Line137-139), as data processing is a step of the surveillance audio-based rainfall estimation (SARE) system. Moreover, our method was evaluated in the RA_XZ dataset, which has been processed by steps in section 2.1. Therefore, we would not retell the story. However, we gave more information about this dataset, just like what we responded point 2.

Point 6 (Experiments): The model training and the Dataprocessing are two different topics.

Response 6: Thank you for your reminder, we have rearranged the Model training separately from the dataset (Please see 3.1 Dataset and 3.2 Model Training in the revised manuscript)

Point 7 (Result): The result should be shown in the seperate chapter.

Response 7: The Result is a separate section in 4. Results (Please see 4. Results in the revised manuscript).

Point 8: Different rainfall type should be defined in the methodology. What does it mean with middle, heavy...

Response 8: We added the description of rainfall levels definition in the revised manuscript (Please see Line 236 and Table 2). Thank you for your kind suggestion.

Point 9: line 268: Why again APNN is abbreviated here?

Response 9: We apologize for the confusion generated by the nomenclature of the model. We have now chosen a clearer nomenclature and replaced it in the revised manuscript (Please see Line 45, Line 341 and Line 371).

Point 10: Why Table 4 show the result in discussion but not in the result section? It should be moved.

Response 10: Table 4 in the previous version, which is table 5 in the revised manuscript, is mainly used to show the influence of fusion of different layers (Please see Line 347-351). To answer your question without confusion, we use Table 5 in the new version to refer it, and so other table mentioned here. We agree that Table 5 can be placed in the result section, merging with Table 3, which could benefit readers to compare our method with different strategies to other methods. However, if so, we also need to move table 6 too. We think that it may cause readers lose in numbers to put all these tables together, and it may result in difficulty for readers to associate text and tables, since the analysis is a little bit far away from the corresponding table/data. Therefore, we place Table 5 in the discussion section.

Point 11 (Conclusion): It is concluded that the rainfall detection has been improved so provide a baseline for the integration of water management. Is it really the fact in comparison to another sensors and methods? Please provide more evidence in the discussion.

Response 11: Thank you for your comments, which is highly appreciated. We have rewritten the conclusion section.

1) Firstly, we showed that our model outperforms others in the SARE system and also demonstrated that spatiotemporal networks, attention mechanisms and multi-scale fusion modules can better represent rainfall events (Please see Line 406-410).

2) Secondly, we indicated that the use of surveillance audio for rainfall observation is not intended to replace other sensors, but to complement the existing rainfall observation network. We demonstrated that surveillance audio is possible to serve as triggers for rapid assessments of rainfall spatiotemporal distribution when city experiences short terms of heavy rainfall. In this regard, the proposed method gives researchers/decision-makers faster insights into how the rainfall effect different area of city. (Please see Line 411-414)

3) Finally, we identified limitations in the existing research. On the one hand, our research is currently still a coarse-grained classification of rainfall levels that donot allow for some scenarios need quantitative rainfall value. Therefore, we believe that the future work should focus on the quantitative research (Please see Line 418-420). On the other hand, we only consider common audio features, such as MFCC, Mel, and Log-Mel. We have not explored whether there are audio features that are more applicable to rainfall in more scenarios (forests, farmland, sea, etc.). Therefore, we believe that the future work should explore the acoustic feature that more applicable to rainfall (Please see Line 420-423).

We sincerely hope that this revised manuscript has addressed all your comments and suggestions. We also asked a professional language editing service to edit the use of English grammar. Once again, thank you very much for your constructive comments and suggestions.

Round 2

Reviewer 2 Report

Thank you for the reply. The paper could be accepted.